# Aquaporins and Ion Channels as Dual Targets in the Design of Novel Glioblastoma Therapeutics to Limit Invasiveness

**DOI:** 10.3390/cancers15030849

**Published:** 2023-01-30

**Authors:** Alanah Varricchio, Andrea J. Yool

**Affiliations:** School of Biomedicine, Institute for Photonics and Advanced Sensing, University of Adelaide, Adelaide, SA 5005, Australia

**Keywords:** brain cancer, motility, invasion, membrane signaling, fluid transport, potassium channels, calcium channels, glutamate receptors, AQPs

## Abstract

**Simple Summary:**

Glioblastoma is a devastating brain tumor that, even with the best available treatments, leads to death for most patients in less than two years after diagnosis, due primarily to its highly invasive nature. Certain membrane signaling proteins (ion channels and water channels) that contribute to cell migration are known to increase in abundance as glioblastoma severity worsens, but have not been explored as possible therapeutic targets. This review evaluates the novel proposal that clinical value might be achieved by simultaneously targeting selected combinations of membrane proteins which show patterns of co-occurrence unique to glioblastoma subtypes. The dual targeting of signaling pathways with pharmacological blockers might exploit cell-specific vulnerabilities in glioblastoma while reducing off-target consequences on normal neurons and glial cells. The optimization of target selection and doses could launch innovative methods to control the spreading of pathological brain tumors, enhancing the success of primary treatments (surgery, radio- and chemotherapy) which comprise current best practice.

**Abstract:**

Current therapies for Glioblastoma multiforme (GBM) focus on eradicating primary tumors using radiotherapy, chemotherapy and surgical resection, but have limited success in controlling the invasive spread of glioma cells into a healthy brain, the major factor driving short survival times for patients post-diagnosis. Transcriptomic analyses of GBM biopsies reveal clusters of membrane signaling proteins that in combination serve as robust prognostic indicators, including aquaporins and ion channels, which are upregulated in GBM and implicated in enhanced glioblastoma motility. Accumulating evidence supports our proposal that the concurrent pharmacological targeting of selected subclasses of aquaporins and ion channels could impede glioblastoma invasiveness by impairing key cellular motility pathways. Optimal sets of channels to be selected as targets for combined therapies could be tailored to the GBM cancer subtype, taking advantage of differences in patterns of expression between channels that are characteristic of GBM subtypes, as well as distinguishing them from non-cancerous brain cells such as neurons and glia. Focusing agents on a unique channel fingerprint in GBM would further allow combined agents to be administered at near threshold doses, potentially reducing off-target toxicity. Adjunct therapies which confine GBM tumors to their primary sites during clinical treatments would offer profound advantages for treatment efficacy.

## 1. Introduction

Cancer remains a leading cause of death worldwide, accounting for approximately 10 million deaths in 2020. The global burden of cancer is steadily increasing, with incidence expected to increase by 47% in the coming decades to reach an estimated 28.4 million cases in 2040. According to the World Health Organization, the most frequently diagnosed types of cancer as of 2020 include breast, lung, colorectal and prostate cancers.

Notably, brain and nervous system cancers accounted for less than 2% of all diagnoses, but caused up to 3% of cancer-related deaths in 2020 [1]. Glioblastoma (also known as glioblastoma multiforme; GBM) is an invasive tumor derived from neuroglial progenitor stem cells. Representing more than 60% of all intracranial tumors, glioblastoma is the most common malignant primary brain tumor [2,3]. With a median survival expectancy of 12 to 14 months following diagnosis [4], the outlook for patients diagnosed with glioblastoma multiforme is grim and has remained so for decades despite increasingly advanced attempts at treatment. Although glioblastoma rarely spreads beyond the brain, the aggressive invasion of glioma cells into healthy brain parenchyma makes glioblastoma multiforme particularly difficult to overcome with current treatment regimens. Further limitations are imposed by the extensive cellular heterogeneity of glioblastoma tumors, which can leave treatment-resistant cells poised to initiate relapse.

Metastasis, in which tumor cells disseminate from primary neoplasms to secondary sites, is responsible for more than 90% of tumor-related deaths [5,6]. The metastatic spread of cancer often results in recurrence during definitive treatment failure. A growing demand for new strategies to inhibit cancer cell invasiveness remains unmet. The hierarchical metastatic cascade offers opportunities for intervention at multiple steps, including angiogenesis, detachment from the primary tumor, degradation of the extracellular matrix and the invasion of dissociated tumor cells into neighboring tissues and vasculature.

## 2. Aquaporins, Ion Channels and Ionotropic Receptors Are Emerging as Therapeutic Targets in GBM

Aquaporins (AQPs) and ion channels are among an array of proteins known to be involved in cellular motility mechanisms, and thus are of interest as target proteins in the development of new treatments for halting tumor spread. Although multiple classes of AQPs and ion channels have been implicated in migration for diverse types of carcinomas in vitro and in vivo, most are yet to be evaluated for glioblastoma. The signaling pathway overlap between normal cells in the brain and GBM is a challenge for imposing interventions without causing unacceptable side effects. We propose that simultaneously modulating both AQPs and ion channels could enhance therapeutic potential. Defining optimal combinations of pharmacological agents to impede glioblastoma motility with minimal cytotoxicity could pioneer new approaches for controlling glioblastoma spread during the concurrent administration of standard treatments. Abnormal aquaporin and ion channel activities have downstream effects that have been linked to progression in malignant tumors. Depending on cancer type, cell invasion involves different aquaporins and ion channels that accelerate tumor angiogenesis, enhance cell volume regulation, regulate cell–cell and cell–matrix adhesions, influence the activity of proteases and extracellular-matrix degrading molecules, and coordinate actin cytoskeletal reorganization [7,8,9,10,11,12,13,14,15,16,17,18,19,20]. Apparently exploiting the broad portfolio of channel properties and tissue-specific distributions, multiple channel classes have been associated with cancer-specific demands for the complex processes of invasion and metastasis. Aquaporin subtypes are selectively upregulated in different cancer types, with roles beyond simple passive water conduits (Table 1).

## 3. Roles of Aquaporins

Aquaporins in the superfamily of major intrinsic proteins (MIPs) comprise 15 mammalian subtypes, including AQPs 0–12 in higher mammals and AQPs −13 and −14 in older lineages [42,43,44,45]. AQPs 0–12 are widely expressed throughout the human body (Figure 1) and particularly abundant in cell types involved in fluid or glycerol transport, including eye lens, renal duct cells, epidermal cells, choroid plexus and astrocytes, adipocytes, hepatocytes and the endothelial cells constituting blood and lymphatic vessels [46,47,48,49,50,51,52,53,54,55,56,57].

AQPs 0, −1, −2, −4, −5, −6 and −8, referred to as classical aquaporins, are known primarily as water channels, but also show permeability to urea, gases, H_2_O_2_, ammonia and charged particles [58,59,60,61,62,63,64,65]. The aquaglyceroporins (AQPs 3, −7, −9 and −10), in addition to mediating water flux, also facilitate the passive transport of glycerol and, in some cases, urea, lactate or H_2_O_2_ [59,66,67,68,69,70,71]. AQP11 and AQP12 are distantly related paralogs of the mammalian aquaporins; AQP11 has been suggested to show water and glycerol permeability [72,73].

### 3.1. AQPs Show a Conserved Structural Theme, but Differ in Key Properties

Aquaporin channels comprise four subunits, each with six transmembrane domains showing well conserved amino acid sequences that carry signature motifs, as well as key differences which enable functional specializations (Figure 2). Water pores are present in each monomer, but tetrameric assembly is a prerequisite for water transport efficiency, structural stability and membrane localization. Tetramers in membranes (Figure 2A) in some AQP classes can aggregate into higher-order arrays stabilized by interactions between N-terminal domains [74,75]. The six α-helices in aquaporin monomers (~30 kD) are connected by loops A to E (Figure 2B). Loops B and E contain the hallmark Asp-Pro-Ala (NPA) sequences which are highly conserved in the MIP superfamily and line the narrow pathway of the subunit water pore (inset 1, top row).

In each monomer, the amphipathic pore connecting the intracellular and extracellular vestibules consists of hydrophilic α-carbonyl groups from the polypeptide backbone, and hydrophobic residues [77]. Asparagine side chains in the NPA motifs oriented into the pore act as hydrogen-bond donors and acceptors to coordinate single-file water transport through the pore, whereas positively charged residues repel proton passage [78]. A narrow region in the extracellular vestibule featuring aromatic (ar) and arginine (R) residues is termed the ar/R constriction (inset 2, top row), which provides substrate selectivity by size exclusion based on the limiting diameter [79]. The partial sequence alignment of human aquaporins (Figure 2C) illustrates the diversity of ar/R sites in terms of amino acid size and hydrophobicity, reflecting the range of solute permeability profiles across the MIP family.

### 3.2. Aquaporins Initially Classified as Strict Water Channels Facilitate Fluxes of Diverse Substrates

AQP0, thought to be exclusively expressed in mammalian eye lens junctions [54], has a low unitary water permeability as compared to AQP1, and is sensitive to external pH and calcium [80,81]. Tyr-23 and Tyr-149 residues flanking the extracellular and intracellular sides of the intrasubunit water pore are conserved in AQP0 across phyla, and thought to constitute a ‘phenolic barrier’ which reduces rates of water transport [82,83]. The residue equivalent to Tyr-149 is absent from aquaporins with high rates of water permeation. However, the residue equivalent to Tyr-23 is present in AQP6 (Figure 2C), which shows low basal rates of water transport [57]. AQP0 reconstituted into lipid bilayers shows ion channel activity characterized by large-conductance single channels with a slight anionic selectivity [84,85,86].

In the same theme, AQP1 also functions as a dual water and ion channel. AQP1, abundantly expressed in the kidney, choroid plexus and endothelia as well as other tissues [56], carries cations through the tetrameric center via a cGMP-gated ion pore that is pharmacologically and functionally distinct from the four monomeric water pores [60,61,62,63,64]. Molecular dynamic simulations confirmed by mutagenesis indicate that cGMP interacts with conserved arginine residues in Loop D to trigger the opening of the central pore, facilitating the permeation of monovalent cations including K^+^, Cs^+^, Na^+^, Li^+^ and TEA^+^ [64,65,79,87,88,89,90,91]. AQP1 upregulation in GBM has been linked to enhanced cell–cell adhesion, cellular aggregation, and actin reorganization [18,92]. The roles of AQP1 in cell volume regulation have been proposed to drive morphological adaptation into elongated spindle shapes allowing glioma cells to penetrate narrow extracellular spaces in the brain [20,93]. Blocking the cation conductance of AQP1 has been shown to slow colon cell migration, an effect that was potentiated by the co-application of an AQP1 water pore blocker, suggesting both channel functions are involved in maximizing cell motility [94]. The possible roles of the dual water and ion channels AQP0 and AQP6 in cell motility mechanisms remain unexplored. AQP6, localized to intracellular vesicles such as in acid-secreting cells of the renal collecting duct [57], has been shown to function as an anion-selective channel, confirmed by the mutation of a lysine residue near the monomeric pore to glutamate, which increased permeability to cations [57]. In *Xenopus* oocytes expressing AQP6, the application of HgCl_2_ increased water permeability and ion currents [57,95], though acidic pH, rather than HgCl_2_, is considered a logical candidate for the physiological activator [96].

In GBM, AQP4 colocalizes with chloride and potassium channels at the leading edges of motile tumor cells to enhance locomotion [19,20], promoting cell volume regulation, integrin trafficking and metalloproteinase (MMP) secretion in response to protein kinase C [19,68,93,97], which also reduces osmotic water permeability by inducing AQP4 internalization [98]. AQP4, highly expressed in the central nervous system, facilitates glymphatic fluid circulation, synaptic plasticity and the regulation of pericellular fluid volumes [99,100,101], and is suggested to contribute to CO_2_ transport [102]. AQP4 in astrocyte end-feet controls water movement across the blood brain barrier, essential for nervous system protection and fluid homeostasis [52,53], as demonstrated by the increased infarct volumes caused by ischemic cerebral edema after the downregulation of AQP4 expression by microRNAs (miRNA-320a and miRNA-130a) [103,104,105].

Other classes of AQPs not known to be linked to GBM include AQPs 2, 5 and 8. AQP2, abundant in the apical membranes of renal collecting duct cells, is regulated by vasopressin, which via cAMP-dependent protein kinase A (PKA) signaling causes the redistribution of AQP4 channels from intracellular vesicles to the apical membrane [106,107]. AQP5, expressed in salivary glands, in corneal, bronchial and pancreatic epithelia, and throughout the respiratory tract [55], is sensitive to extracellular signaling-regulated kinase (ERK) and cAMP signaling pathways [108,109,110]. In addition to mediating transmembrane water and CO_2_ fluxes [102,111,112], AQP5 has been proposed to provide H_2_O_2_ transport. When expressed in yeast cells, human AQP5 enabled H_2_O_2_ and reactive oxygen species (ROS) regulation not evident in AQP5-null control cells [113], and influenced microtubule dynamics and paracellular passage of molecules in subepithelial tissues [55]. AQP8 expressed in colonic and pancreatic tissues [49] relocates between the membrane and cytoplasm in response to osmotic load [114], and shows a broad solute permeability profile including NH_3_, NH_4_^+^ and H_2_O_2_ as well as water. The heterologous expression of AQP8 in a mutant yeast strain deficient in NH_3_ transport restored cell growth [115]. In voltage-clamp work using *Xenopus* oocytes, AQP8 expression enabled inward currents carried by NH_4_^+^ [116], suggesting ion channel functionality. The heterologous expression of AQP8 also increased capacity for H_2_O_2_ diffusion in yeast and HeLa cells [62,117]. AQP8 in the inner mitochondrial membrane is thought to mediate H_2_O_2_ release from the mitochondrial matrix when ROS generation is high [118]. Since the initiation and progression of gliomas has been shown to be associated with cellular redox imbalances [119], further investigation of the possible roles of AQPs as peroxiporins in GBM would be of interest.

The diverse subtype-specific capabilities of AQPs appear to be differentially exploited by cancers for enhanced progression and survival. The AQP classes linked to the pathologies are not interchangeable, suggesting that selective pharmacological modulators could prove useful as targeted treatments which could be tailored to specific types of cancers based on the corresponding classes of AQPs involved.

### 3.3. Aquaglyceroporins Enable Transmembrane Fluxes of Water and Glycerol, and Other Substrates

A comparison between the ar/R constriction regions of classic aquaporins such as AQP1 and the aquaglyceroporins AQPs 3, 7, 9 and 10 (Figure 2C) suggests that a histidine residue in classic AQPs might serve to exclude glycerol from the water pore [120,121] since the equivalent position in aquaglyceroporins features small, non-polar amino acids, consistent with a wider pore for glycerol passage [121]. Of the aquaglyceroporins, only AQP9 has thus far been linked to invasion properties in GBM. AQP9 features one of the most diverse permeability profiles of the mammalian MIP family; proposed substrates include H_2_O_2_, mannitol, sorbitol, adenine, uracil and thiourea [122], with possible roles in absorption and excretion that await further investigation. It is postulated that the AQP9-mediated clearance of lactate and glycerol facilitates glioblastoma survival by counteracting the lactic acidosis which occurs during extensive hypoxia within glioblastoma tumors [123,124].

Not linked to GBM are AQPs 3, 7 and 10, which also show tissue-specific roles in glycerol handling. AQP3 is found in the kidney medulla, colon and keratinocytes in the basal layer of the epidermis [47]. AQP3-facilitated glycerol transport is vital for skin hydration, elasticity, and wound healing [47,125]. In cultured human keratinocytes, reduced AQP3 protein expression caused dehydration, reversed by a glycerol derivative that upregulated AQP3 activity and reduced transepidermal water loss [126,127]. AQP3 also shows permeability to urea, potentially relevant to the function of kidney collecting duct cells [128]. AQP7 mediates the efflux of newly synthesized glycerol during lipolysis in adipose tissue, and contributes to adenosine triphosphate (ATP) production in cardiomyocytes [48,51]. AQP7-null mice showed glycerol accumulation in adipocytes, increased triacylglycerol synthesis, and ultimately obesity with severe insulin resistance [129,130]. AQP7, unlike most other mammalian aquaporins, has atypical signature motifs (NAA and NPS instead of NPA), as shown in Figure 2C [131,132]. Colocalized with AQP1 in the apical membranes of proximal tubules [66,133,134], AQP7 supports kidney glycerol reabsorption [134]. The removal of NH_3_ from the circulation to reduce toxic accumulation in arterial blood plasma during intense exercise [135,136] could involve AQP7 or AQP9, both of which mediate NH_3_ permeability as shown in *Xenopus* oocytes at levels exceeding that of AQP3, and not seen for AQPs 2 or 4 [128,137]. In hepatocytes, AQP9 facilitates sinusoidal glycerol uptake from the bloodstream for gluconeogenesis during starvation [46]. In addition to water and glycerol, both AQP7 and AQP9 show permeability to urea, NH_3_ and arsenite. The transport of arsenite by AQPs 7 and 9 could implicate these aquaglyceroporins in aiding or abetting responses to toxic substances [138]. Reduced blood glucose levels in fasted AQP9-null mice reflected impaired gluconeogenesis [139]. However, possible correlations between human aquaporin levels, obesity and diabetes remain controversial. AQP10 is highly expressed in villi of the proximal small intestine, and in adipocytes with AQPs −3 and −9, which promote glycerol export [50]. AQP10 shows water, glycerol and urea permeability [140] and pH-sensitive glycerol transport [141]. Double protonation of a key histidine residue at low pH is linked to structural rearrangement of the ar/R constriction, resulting in pore widening and glycerol passage. In contrast, water flux through AQP10 is pH-insensitive [142].

AQPs 11 and −12 have been dubbed superaquaporins, with an overall structural architecture characteristic of AQPs [44,143], but with relatively low sequence similarity as compared to other MIP classes [44,143]. NPC and NPT are seen instead of the signature motif NPA [44] (Figure 2C). Physiological roles remain to be undefined, although glycerol permeability has been reported for AQP11, which might contribute to salivary gland development and intravesicular homeostasis [144,145]. AQP12 in pancreatic cells could influence the secretion of digestive enzymes and fluids [146] but details are not yet clear.

### 3.4. Aquaporins as Targets for Therapeutic Treatments in GBM

Aquaporin classes have been well characterized in terms of protein structure, function and localization, but their molecular mechanisms in tumorigenesis, metastasis, cellular migration and invasion remain to be fully defined. Diverse roles in pathologies including nephrogenic diabetes, cerebral edema and dry eye disease [147,148,149] suggest great potential for aquaporin-based therapeutics. However, single targets addressed with single agents might not achieve the therapeutic goals needed in complex systems such as the CNS. Combinations of pharmacological agents might offer greater hope for successful interventions.

The field of aquaporin pharmacology is still at an early stage. Unlike ion channel blockers, a broad panel of defined aquaporin inhibitors is not yet available for subtype-specific modulation [150]. Blockers of AQP1 that slow colon cancer migration and invasion include bumetanide derivatives AqB007 and AqB011, which inhibit the AQP1 cation channel; 5-hydroxylmethyl furfural, which also blocks the AQP1 ion channel; and Bacopasides I and II from the medicinal plant *Bacopa monnieri*, which preferentially block the AQP1 ion channel and water pores, respectively [151]. Treatment with Bacopasides I and II together potentiates the migration-inhibiting effect [152], suggesting that the dual inhibition of AQP1-mediated water and ion movement is more efficacious than the inhibition of either water or ion channel activity alone.

## 4. Ion Channels as Targets of Interest for Controlling GBM Progression

Transcriptomic analyses have identified classes of ion channels enriched in human glioblastoma biopsy samples [153,154] and that are involved in cancer invasion and metastasis in diverse cancer types (Table 2). These channels could be targeted perhaps in combination with aquaporin blockers to restrain tumor progression.

Much of the work on ion channels in cancers to date has been defining subtypes which influence cancer cell growth and survival as strategic targets for the design of small molecule inhibitors to disrupt proliferation or induce apoptosis [176]. An equally important area that merits investment is defining the ion channels involved in cancer cell migration and invasion, to identify avenues for the co-development of small molecule inhibitors of motility. Diverse classes of ion channels are gated by an array of physiologically relevant parameters including voltage, ligand binding, pH, membrane tension, and more, which trigger conformational changes to open, close, potentiate or inactivate pores for ion conduction [177] and to control key cellular responses. The highly invasive nature of GBM tumors constitutes a major challenge for effective treatment using strategies, such as chemotherapy and radiotherapy, which aim to directly eradicate primary tumors. The selective targeting of ion channel subtypes that are enriched in GBM tumors and known to influence cellular motility pathways could constitute a powerful adjunct therapy. A prior review surveyed the association between increased expression of KCa, Ca2+, Na+, GABA and ACh receptor ion channels and reduced survival in GBM patients, with a view towards identifying points for anti-invasive interventions [178]. Updated information is included in Section 4.1 and Section 4.2 below.

### 4.1. Roles for K^+^ and Ca^2+^ Channels in GBM

K^+^ and Ca^2+^ channel activities influence motility in GBM and other cancers. The association between small-conductance K_Ca_ channel K_Ca_2.3 and the risk of bone metastasis by human breast cancer cells (MDA-MB-435s) was shown to depend on formation of a lipid-raft complex with the store-operated Ca^2+^ channel Orai1 to promote migration [179]. Intracellular Ca^2+^ oscillations modulate cellular migration in U87-MG tumor cells by governing the turnover rate of focal adhesions, the proteins that anchor tumor cells to the ECM [180]. Knockdown of the intermediate-conductance K_Ca_ channel K_Ca_3.1 by short-hairpin-RNA abolishes chemokine-dependent cell migration in both primary GBM cells and glioblastoma cell lines [181]. Apamin, a selective inhibitor of small-conductance K_Ca_ channels [182], emerges as a candidate of particular interest for controlling GBM motility given its ability to cross the blood–brain barrier [183]. Apamin does not appear to have been tested previously in GBM (pubmed.ncbi.nlm.nih.gov); (accessed on 20 January 2023), but shows promise in recent work (Varricchio et al., 2023, MS in review). Observations of the inhibitory effects of nifedipine and apamin on motility in a variety of other tumor types [184,185,186] have set a useful precedent for the idea that K^+^ and Ca^2+^ ion channel modulators might also be useful for controlling motility in GBM.

### 4.2. Roles for Acid-Sensing Ion Channels and Volume-Regulated Anion Channels in GBM

Recent work has suggested that acid-sensing ion channels (ASICS) and volume-regulated anion channels (VRACs) contribute to the growth and motility of GBM. This is not surprising given that: (i) an acidic tumor microenvironment (pH 5.6 to 6.8) is a hallmark of malignant tumors cells such as GBM [187]; and (ii) GBM cells tightly regulate their volume to adapt to the spatial constraints imposed by the narrow extracellular spaces of the brain [93]. In GBM cell lines A172 and U87-MG, weakly acidic conditions emulating a tumor microenvironment promoted migration and invasion by up to 67%. This effect was attributed to the activation of ASIC1, based on results showing that the selective ASIC1 inhibitor psalmotoxin abolished glioma cell migration within cell monolayers and prevented GBM cell invasion through an ECM-like membrane [188]. Similarly, decreased ASIC1 protein expression in ASIC1-siRNA-treated U87-MG cells was associated with reduced migration in weakly acidic conditions without impairing proliferation [188]. Further studies with patient-derived glioma cells using similar techniques could assist in delineating the role of ASIC1 as a therapeutic target across multiple classes of glioblastomas.

Ubiquitously expressed VRACs have been implicated in tumor cell proliferation, migration and apoptosis [189]. VRACs are heteromeric channels comprising leucine-rich-repeat-containing protein 8A (LRRC8A) with at least one other LRRC8 isoform [190,191,192], and are activated by extracellular hypertonicity. VRACs mediate the permeation of chloride and, in some cases, organic anionic osmolytes, altering osmotic gradients to drive water flux and regulating cell volume [191,193]. Interestingly, VRACs in chronic myelogenous leukemia cells have been found to serve more than one role, mediating the uptake of cisplatin as well as facilitating apoptosis [194]. The role of VRAC channels in GBM is less clear. A study on GBM cell lines U87-MG and U251-MG found a block of swelling-activated Cl^–^ channels with DCPIB suppressed migration and invasion [195]. In contrast, subsequent work reported that neither DCPIB nor the siRNA-mediated knockdown of LRRC8A affected migration in U87-MG or U251-MG [196]. Defining variables that might influence the levels of VRAC contributions to the migration and invasion of GBM tumor cells awaits further investigation. Collectively, these findings suggest that ASICs and VRACs merit consideration as therapeutic targets for the treatment of GBM, though details remain to be confirmed and demonstration of translational value in vivo for these and other ion channels remains a gap in knowledge.

## 5. Ligand-Gated Channels as Pharmacological Targets in GBM Cancer Progression

Signal transduction from chemical (neurotransmitter) to electrical (voltage) is accomplished by channel-mediated ion fluxes generating postsynaptic potentials [197]. One of the major superfamilies of ionotropic receptors features pentameric assemblies of subunits around a central ion pore. This pentameric superfamily includes nicotinic acetylcholine receptors, γ-aminobutyric acid (GABA) receptors, glycine receptors, serotonin receptors and others. Neuronal nicotinic acetylcholine and GABA_A_ receptors have been identified in GBM cells [198,199]. In glioblastoma cell lines, the activation of neuronal nicotinic acetylcholine receptors (nAChRs), cation-selective channels which mediate fast neurotransmission [200,201], has been associated with increased invasion [202].

In addition to L-type Ca_V_ channels, K_V_ channels and K_Ca_ channels, AMPA/kainate receptors and GABA_A_ receptors have been implicated in mechanisms of cellular motility. Table 3 summarizes effects on tumor cell invasion and migration in vitro with pharmacological inhibitors of some of the ion channels known to be upregulated in GBM, including AMPA/kainate receptors, GABA_A_ receptors and K^+^ and Ca^2+^ channels.

Patterns of upregulated expression in ionotropic glutamate receptors in GBM are a focus of keen interest for understanding the mechanisms of cancer progression [178]. The glutamate receptor family built as tetramers includes subtypes classified as NMDA (N-methyl-D-aspartate) and non-NMDA, subdivided into AMPA (α-amino-3-hydroxy-5-methyl-4-isoxazolepropionic acid) and kainate [206,207] depending on the preferred activating ligands [208,209]. AMPA/kainate receptors are inherently permeable to Na^+^ and K^+^; an additional permeability to Ca^2+^ is present in AMPA receptors which lack the GluA2 subunit [210]. In GBM, increasing the expression of the GluR2 by adenoviral transfection converted AMPA receptors to a Ca^2+^-impermeable phenotype and resulted in increased apoptosis and reduced migration, showing that Ca^2+^ influx through glutamate receptors is important for the glutamate-induced stimulation of motility [174].

In GBM cells xenografted into mouse hippocampuses, voltage-clamp recordings revealed stimulation-evoked fast inward currents consistent with excitatory postsynaptic currents, without the fast large-amplitude currents which drive action potentials. These results supported the conclusion that *bona fide* synapses form between presynaptic neurons and postsynaptic subpopulations of GBM cells, and that synaptic responses in the glioma cells are dependent on neuronal firing patterns [211]. Neuronal activity directly regulated glioma membrane potential via neuron-to-glioma synaptic transmission involving glutamate receptors, creating depolarization responses in turn linked to glioma proliferation and invasion in vivo [211]. Excitatory postsynaptic responses measured in a subpopulation of glioma cells by voltage clamp were shown to be mediated by postsynaptic AMPA receptors, in response to glutamate presynaptically released from electrically active neurons. A subset of glioma cells with prolonged currents not matched temporally with neuronal synaptic transmission were nonetheless found to be sensitive to the action potential blocker tetrodotoxin, suggesting that responses in these glioma cells were aligned with slower astrocyte responses to neuronal activity [211], reflecting glial glutamate transporters and inward rectifiers used for removing excess extracellular potassium and glutamate released from neurons during rapid firing [212]. The ability of GBM cells to acquire postsynaptic elements of neurotransmission provides an intriguing insight into the role of the surrounding neuronal network in governing and enabling cancer progression.

## 6. Synopsis of Candidate Signaling Pathways in GBM

Regulation of cell membrane potential involves a dynamic balance of ion channels, transporters, ionotropic receptors for the control of signaling pathways [213], illustrated in Figure 3.

Membrane potential can affect multiple aspects of cancer cell motility, in part by regulating Ca^2+^ channel activity and shifting the the driving force for Ca^2+^ entry [214,215] in a dynamic process. A more negative membrane potential will increase Ca^2+^ influx, if channels are open. Membrane depolarization events enable Ca_V_ channel activation, but prolonged depolarizations cause inactivation [215]. During cell migration, fluctuations in Ca^2+^ concentration regulate tractional forces, directional sensing, cytoskeleton reorganization, and more [14,216]. Beyond the well-defined action potential which serves as the classic signaling event in excitable cells, the baseline membrane potential is a key biophysical regulator in all cells for proliferation, differentiation, cell-volume control and other processes [217,218].

Hundreds of ion channels encoded in the mammalian genome are specialized into multiple isoforms that cover a broad spectrum of properties, allowing the tailoring of channel subtypes into precise physiological roles meeting cell-specific needs, and thus prompting keen interest in this class of proteins as drug targets. The expression of excitatory ion channels such as ASICs, nAChRs and GluRs in GBM cells appears to exacerbate GBM invasiveness; however, overdriving the membrane depolarization phase is deleterious to GBM (as it is in other cells), as was shown using pharmacological blockers of K_Ca_ channels, which induced cell death in glioma-initiating cells and improved the survival of mice in an orthotopic model of human GBM [219]. Handicapping K_Ca_ channels and GABA_A_Rs to interfere with repolarization [220,221,222] would be an intriguing area to explore in future work, identifying more targets for candidate GBM therapies.

Other membrane channels in GBM also contribute to signals controlling migration (Figure 3). ASICs allow Na^+^ permeation, generating depolarizing inward currents in response to external acidification [223,224]. Acetylcholine binding to nAChRs at the interface of adjacent subunits induces conformational changes which open the pore to permit cation permeation, predominantly Na^+^ entry (with subtype-specific permeation by Ca^2+^) to evoke depolarization [201]. Similarly glutamate binding to AMPA/kainate receptors enables cation currents, with predominantly Na^+^ influx (and subtype-specific Ca^2+^) driving depolarization. Activation of Ca_V_ channels by depolarization, or action of Ca^2+^ entry through permeable ligand-gated receptors, in turn can activate K_Ca_ channels in a negative feedback loop that repolarizes the membrane [222]. Inward rectifier K_ir_ channels assist in maintaining a negative resting membrane potential by holding the membrane potential closer to the equilibrium potential for K^+^ [225]. Mechanisms for both depolarization and repolarization are necessary for cancer progression.

GABA_A_ receptors are inhibitory, featuring a central pore permeable to Cl^−^ and other anions such as HCO_3_^−^ [226,227], which in response to agonist binding can facilitate net anion influx and hyperpolarization [220,221]. Glioma cells upregulate GABA_A_ receptor expression in vitro and in vivo following contact with neurons; the activation of GABA_A_ receptors inhibits proliferation in glioma [228]. An interesting avenue to be explored would be testing effects of GABA agonists and antagonists on GBM invasiveness.

Simultaneous pharmacological blocks of aquaporin and AMPA/kainate receptors could present a powerful approach for slowing invasive cell spread, based on recent findings in GBM cell lines (Varricchio et al., 2022, in review) showing the augmented effects of AMPA/kainate channel inhibitors when combined with AqB013, a bumetanide derivative which blocks water passage through AQP1 and AQP4 [229]. The mechanisms by which aquaporins and ion channels regulate cellular motility in cancers warrant investigation in the identification of pharmacological blockers and the design of novel glioblastoma treatments directed towards halting invasion, but are likely to require targeting more than one signaling pathway to realize clinically relevant outcomes.

## 7. Conclusions

Clusters of membrane signaling proteins including aquaporins and ion channels are upregulated in GBM, and here are proposed as attractive targets for novel methods to reduce glioblastoma motility, aimed at confining GBM tumors to primary sites of origin. Pending the development of a complete pharmacological portfolio for all the candidate channel classes, testing combined inhibitors of aquaporins and ion channels could aid in narrowing the effects of treatments to be specific to GBM. Transcriptomic analyses of GBM patient biopsies have shown that the prognostic impacts associated with levels of expression are stronger for sets of channel genes than for individual genes alone [178]. This observation highlights the concept that cellular signaling is carried out by sets of proteins with partially overlapping functional properties, providing a level of redundancy that enables compensatory responses.

The pharmacological inhibition of aquaporin and ion channel classes has been used to reduce tumor cell motility in other cancer types, but needs to be tested in GBM and extended into in vivo models to address potential translational benefits. Optimal combinations of pharmacological agents which synergistically act to halt glioblastoma motility with minimal cytotoxic side-effects could offer new promise for designing low dose therapies that effectively limit glioblastoma invasion, ideally without disrupting normal neural networks. Novel combinations of inhibitors used as adjunct therapies could aid success by extending the durations of effective windows for the administration of front line clinical treatments.

## Figures and Tables

**Figure 1 cancers-15-00849-f001:**
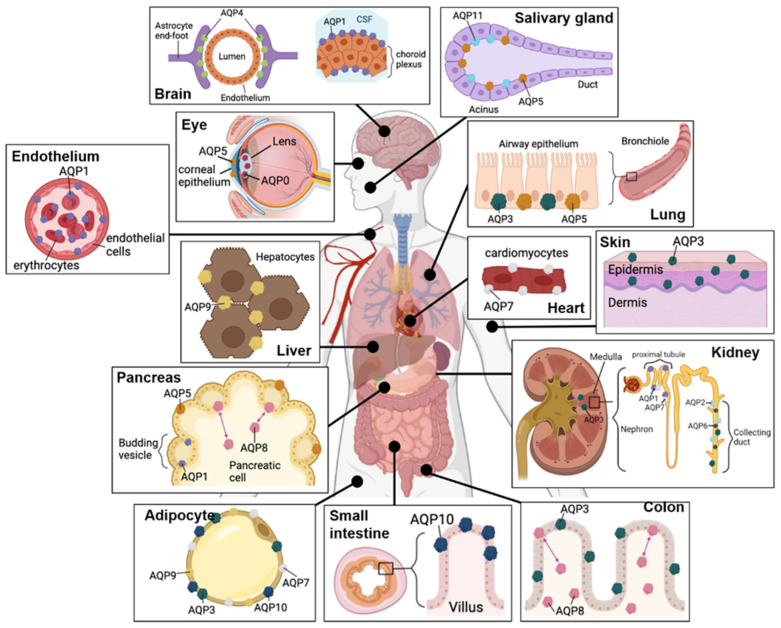
Overview of mammalian aquaporin tissue distributions. In the brain, AQP1 is abundant in the choroid plexus, and AQP4 is high in astrocyte end-feet. AQP1 is the predominant aquaporin in erythrocytes and blood vessel endothelial cells. AQP5 and AQP11 are expressed in the salivary glands. In the eye, AQP5 is expressed in the epithelial cells of the cornea, and AQP0 in the lens. AQPs 3 and −5 are expressed in the epithelium of the respiratory tract. Hepatocytes, cardiomyocytes and epidermal keratinocytes express AQPs 9, −7 and −3, respectively. In pancreatic cells, AQPs 1 and −5 are expressed on the membrane; AQP8 shuttles between intracellular and membrane locations. In the kidney, AQP3 is expressed in the medulla and collecting ducts; AQPs 1 and −7 are colocalized in the proximal tubule; AQPs 2 and −6 are expressed in the apical membranes of the collecting ducts. Adipocytes express aquaglyceroporins AQPs −3, −7, −9, and −10. AQP10 is highly expressed in intestinal villi. AQP3 and AQP8 are abundant in colonic tissue.

**Figure 2 cancers-15-00849-f002:**
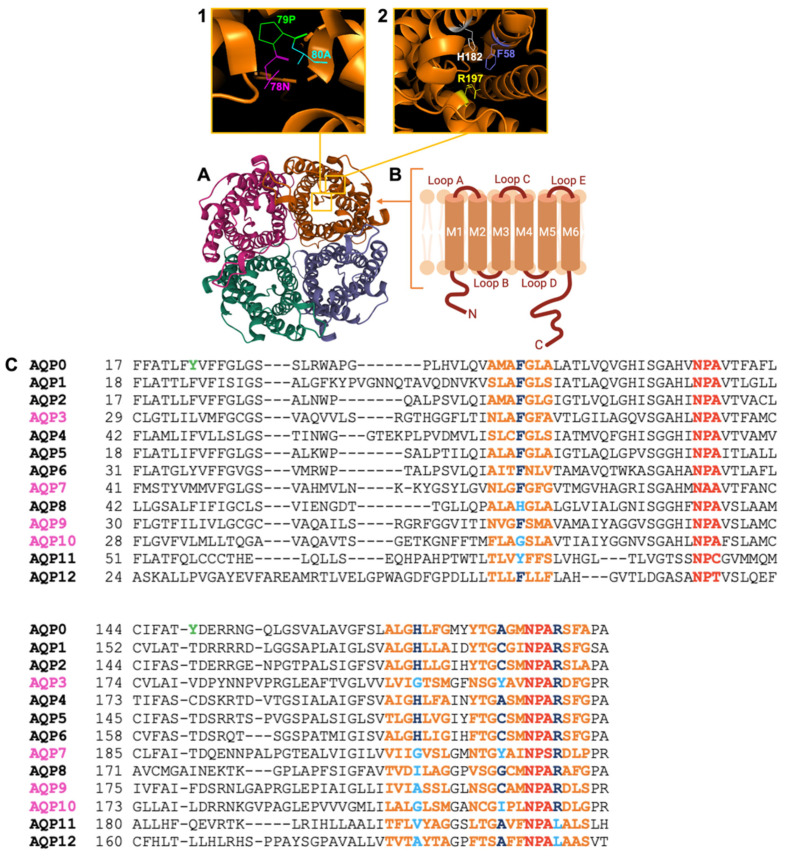
Schematic representation of the Aquaporin-1 channel structure. (**A**) The extracellular face of tetrameric AQP1, from the X-ray crystal structure for bovine AQP1 (Protein Data Bank 1J4N). (**B**) Simplified transmembrane topography diagram of an individual subunit showing six helical transmembrane domains (M1–M6) connected by loops A–E. (**Inset 1**) View of Pro 79, Asp 80 and Asn 78 residues in the signature NPA motif. (**Inset 2**) View of His 182, Phe 58 and Arg 197 residues in the ar/R constriction enabling selective water permeability. (**C**) Partial amino acid sequence alignment of human AQPs 0–12 comparing the NPA motifs (red) and ar/R constriction regions (orange). Aquaglyceroporins are shown in magenta. Conserved residues within the ar/R constrictions are highlighted in navy; deviations from this are highlighted in cyan. The Tyr phenolic barrier specific to AQP0 is shown in green. Panels A and B are adapted from Varricchio et al. [76].

**Figure 3 cancers-15-00849-f003:**
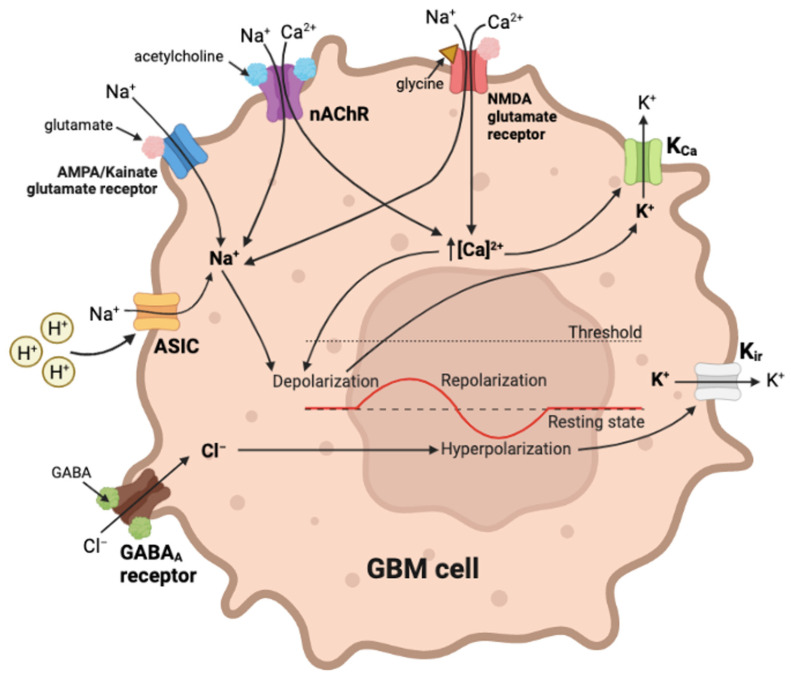
Multiple classes of channels and receptors are expressed in GBM cells; a subset are shown here for illustration. AMPA/kainate-and NMDA-type glutamate receptors carry net inward Na^+^ currents, and a subset also facilitates Ca^2+^ influx. Nicotinic ACh receptors (nAChR) induce depolarizing Na^+^ (and subtype-dependent Ca^2+^) influxes. ASICs open in response to acidic extracellular conditions, allowing Na^+^ influx. Intracellular Ca^2+^ activates K_Ca_ channels, driving repolarization. Inhibitory GABA_A_ receptors enable a hyperpolarising net influx of Cl^−^. K_ir_ channels to maintain the negative resting membrane potential.

**Table 1 cancers-15-00849-t001:** Proposed roles for aquaporins and aquaglyceroporins in elevating invasion and metastasis in different types of cancers, with cited references in brackets.

Channel Type	Cancer Types [Reference]	Roles in Cancer Invasion and Metastasis
AQP1	Glioma [21] Lung adenocarcinoma [22] Colorectal carcinoma [23] Multiple myeloma [24]	Enhancing tumor endothelial cell migration, promoting the recruitment of new tumor vasculature Regulating tumor cell volume and interactions with the cytoskeleton to enable cellular protrusion formation, promoting migration and invasion Interacting with metalloproteinases to degrade the extracellular matrix (ECM) and enable the invasion of tumor cells into neighbor tissue Interacting with β-catenin to enhance migration
AQP2	Endometrial carcinoma [25]	Regulating focal adhesion turnover, facilitating the traction of invading tumor cells to ECM Reorganizing F-actin for the estradiol-induced invasion and adhesion of tumor cells
AQP3	Lung cancer [26] Gastric cancer [27] Liver and pancreatic cancer [28,29] Colorectal carcinoma [23] Ovarian cancer [30] Breast cancer [31]	Interacting with ECM-degrading proteases Regulating cell protrusion formation, enhancing tumor cell migration Enabling the H_2_O_2_ influx preceding chemokine-dependent tumor cell migration
AQP4	Glioma [21]	Colocalizing with ion channels at leading and trailing edges of migrating tumor cells Regulating tumor cell volume and cytoskeletal dynamics to enable cellular protrusion formation for rapid invasion and migration
AQP5	Prostate cancer [32] Myelogenous leukemia [33] Liver cancer [29] Pancreatic cancer [28]	Colocalizing with ion channels at the leading and trailing edges of migrating tumor cells Enabling rapid cellular protrusion formation via cell volume modifications
AQP7	Thyroid cancer [34]	Unknown
AQP8	Oesophageal cancer [35] Myelogenous leukaemia [36] Cervical cancer [37]	Enhancing tumor cell migration via the EGFR/ERK1/2 pathway following EGF stimulation Facilitating the influx of Nox-derived H2O2, increasing intracellular ROS to levels that drive tumor cell proliferation and migration
AQP9	Glioblastoma [38] Astrocytoma [39] Prostate cancer [40]	Potentially accelerating the ERK1/2 pathway ECM-degrading metalloproteinase-9
AQP10	Gastric cancer [41]	Unknown; mRNA upregulation associated with poor prognosis

**Table 2 cancers-15-00849-t002:** Proposed roles for ion channels in elevating invasion and metastasis in different types of cancers, with cited references in brackets.

Channel/Receptor Type	Cancer Types Showing Upregulation [Reference]	Roles in Cancer Invasion and Metastasis
Ca_V_	Breast cancer adenocarcinoma [155] Head and neck carcinoma [156]	Promoting filopodia stabilization, allowing maturation into focal adhesions that direct cancer cell migration and invasion Facilitating EGFR signaling and ECM stiffening to induce collective cancer cell invasion of the surrounding tumor microenvironment
Na_V_	Breast cancer [157]	Altering cytoskeletal elements to adopt cellular morphologies facilitative of migration Promoting proteolytic degradation of the ECM
K_V_	Melanoma [158,159] Neuroblastoma [160] Breast cancer [161]	Interacting with β1-integrins to induce the membrane hyperpolarization preceding extensive cell dispersion and actin cytoskeleton reorganization Modulating cell migration by allowing calcium entry and inducing rapid focal adhesion turnover rates
GABA_A_R	Breast cancer [162,163] Lung cancer [164]	Activating Akt and ERK1/2 kinases to facilitate downstream cellular migration pathways Mediating upregulation of ECM-degrading metalloproteins
nAChR	Breast carcinoma [165] Cervical cancer [166]	Facilitating nicotine- and growth factor-induced increases in redox regulator thioredoxin, overexpression of which induces: Increased production of VEGF, HIF-1a and angioproteins; potent stimulators of cell migration Rho-mediated cytoskeletal remodeling
K_ir_	Gastric cancer [167]	Interacting with ser/thr kinase 38 (Stk38) to enhance MEKK2-MEK1/2-ERK1/2 signaling, promoting invasion and metastasis
K_Ca_	Glioblastoma [20,168,169] Melanoma [159]	Allowing potassium fluxes across the membrane that accompany osmotically driven water movements, allowing changes in cell volume and shape facilitative of cell locomotion Modulating actin polymerization and depolymerization in response to cellular volume fluctuations
ASIC	Breast cancer [170] Epithelial carcinoma [171] Pancreatic cancer [172]	Facilitating increased calcium influx under acidic conditions to promote: ROS-AKT-NF-κB signaling that regulates cell migration and invasion Epithelial-to-mesenchymal transition via the RhoA pathway, during which tumor cells acquire invasive capabilities
NMDA receptor	Lung and thyroid carcinomas, medulloblastoma [173]	Modulating Ca^2+^ homeostasis to regulate cellular process formation and migration
AMPA/kainate receptor	Glioblastoma [174] Pancreatic cancer [175]	Facilitating the Ca^2+^ entry required for cells to adopt fusiform morphology and elongated processes suitable for cell locomotion Inducing cellular invasion and migration via activation of the K-ras/MAPK signaling cascade in response to increased glutamate levels

**Table 3 cancers-15-00849-t003:** Effects of selective ion channel inhibition on cellular motility in in vitro cancer models.

Protein (*Gene*) of Interest	Pharmacological or Genetic Inhibitor	Effect(s)
AMPA/kainite receptor GluR-2 (*GRIA2*)	Cyanquixaline (CNQX) 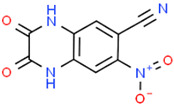	Decreased invasion-promoting neuron-glioma synaptic signaling [165].
L-type voltage-gated Ca^2+^ channels Ca_V_1.1 (*CACNA1C*)	Nifedipine 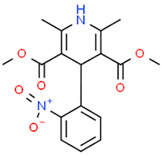	Suppressed invasion and migration of tumor cells sourced from primary colorectal cancer specimens [184].
Voltage-gated K^+^ channel K_V_1.1 (*KCNA1*)	4-aminopyridine (4-AP) 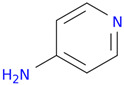	Orthotopic xenograft mouse models of pediatric GBM [166]
Small-conductance Ca^2+^-activated K^+^ channels K_Ca_2.2 (*KCNN2*) K_Ca_2.3 (*KCNN3*)	Apamin 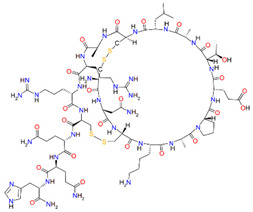	Inhibited invasion in breast cancer carcinoma cell lines MDA-MB-435s cells and MCF-7 [185]. Decreased migration in: MDA-MB-435s cells expressing K_Ca_2.2- or K_Ca_2.3 [203,204]. Melanoma cell lines Bris and 518A2 [186].
GABA_A_ receptors (*GABRA1*)	microRNA miR-139-5p	Inhibited migration and invasion in GBM cell lines U87-MG and U251-MG [205].

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
