# Peer review of "Aquaporins and Ion Channels as Dual Targets in the Design of Novel Glioblastoma Therapeutics to Limit Invasiveness"

_cancers, 2023, doi:10.3390/cancers15030849_

Round 1
Reviewer 1 Report
The authors have evaluated numerous aquaporins and ion channels and proposed their effect on reducing the motility of glioblastoma multiforme, a common variant of malignant brain tumor. With detailed figures and tables, the author have explained the roles and structural features of aquaporins including aquaglyceroporins and ion channels by reviewing invasive and metastasis cancer-associated aquaporins and ion channels, aquaporin expression patterns throughout human body and partial sequence alignment studies. This provided insights on the unique ion permeability feature in the major intrinsic proteins superfamily, including aquaporins. Similarly, ion channels are explored as potential targets due to their effects on cancer progression. In addition to these, the authors have also evaluated potential aquaporins and ion channels not yet directly linked to GBM progression. For instance, the possible role of aquaporins in association with GBM is via extracellular signalling pathways and cellular redox imbalances, whereas ion channels employ pH sensing as a means to modulate tumor progression.
The review article is promising for its future readers in terms of its straightforward writing and the focus of the review topic.
Overall the authors have done very good work on framing the review structure, the figures and schemes are self-explanatory and well-represented.
Author Response
We thank the reviewer kindly for the positive evaluation of the manuscript.
Reviewer 2 Report
Yool and her colleague provide an excellent review on Aquaporins and Ion Channels in the development of glioblastoma therapeutics with Reduced Invasiveness. In the first section, the authors discuss about downstream consequences of abnormal aquaporin and ion channel activity associated to glioma development. Following this, aquaporins and ion channel’s functional consequences were discussed . In particular section 3 and 4 provides detailed information about sequence, structural, and expression profiles of aquaporins and ion channels in signaling pathways involved in glioma. The paper is well written, the data presented supports the conclusion (title of the paper). This is an important contribution to the field of cancer biology and an excellent addition to the literature. I recommend the paper for publication.
Minor comments to be addressed
Replace a-helices with α-helices (page 4, line 135)
The text in some figures seems to be blurred. Provide high resolution figures (particularly Figure 4).
Author Response
We would like to thank the reviewer for the positive review and for catching important details. Concerns raised are addressed below:
1. Replace a-helices with α-helices (page 4, line 135)
We thank the reviewer for catching this typographic error. We suspect this reformatting occurred when converting the original document into the Palantino font of the Cancers template. Each α is now inserted as a symbol instead of using the ‘Symbol’ font.
2. The text in some figures seems to be blurred. Provide high resolution figures (particularly Figure 4).
High quality PDF images have been incorporated for each of the figures used in the revised MS. The former Figure 4 was removed entirely to minimize the overly general information on ion channels, and focus the presentation more clearly on classes involved in GBM motility pathways.
Reviewer 3 Report
In this review, the authors give a summary of the current knowledge on the roles of aquaporins and ion channels in glioblastoma biology (mainly metastasis) and therapy. The part about aquaporins is well-written, clear and nearly comprehensive. The part on ion channels, however, is rather superficial in its trial to cover everything. With that, there are a number of unclarities if not simply errors. Parts on the ion channels should be clearly improved. In some parts it feels like the authors want to repeat and combine their previous reviews without setting a clear focus and neglecting some new developments.
In the following some comments and suggestions that might improve the review:
1) There is quite an overlap with a recent review by the same authors in the IJMS. The authors should make clear what the new direction of the present review is.
2) Why Table 1A and 1B instead of Table 1 and Table 2?
3) Lines 308f: “Transcriptomic analyses have identified classes of ion channels enriched in human 308 glioblastoma biopsy samples [79,154]” The authors should cite original research on the transcriptomics here rather than their own previous review/hypothesis paper – this is a general concern about the review: it often cites other reviews instead of the actual studies that showed (or did not) what the authors state.
4) Line 311: There is more than voltage- and ligand-gated channels that are investigated, e.g. acid-sensitive (protonation is not a ligand), mechanosensitive, volume-regulated. These are not studied in respect to glioblastoma?
5) Table 1B: what does ClC mean? Ref 20 is on ClC-3, which by now is known to be an endosomal Cl/H exchanger, no Cl channel of the plasma membrane. Therefore, unlike stated at various places in this review, ClC-3 cannot mediate Cl efflux for cell shrinkage. Instead, although ClC-3 briefly was a candidate for the volume-regulated anion channel, its identity has been revealed as LRRC8 proteins (see below). Ref 163 does not show anything on ClC. This part on chloride channels and exchangers is very superficial and unclear. Which CLC was actually shown to play what role in which process?
6) Figure 4D does not make sense, the ion and water flow are mixed up. Water efflux does not parallel ion influx during RVD (possibly Cl in neurons, but then there needs to be efflux of other osmolytes that drives water efflux). Again, ref. 20 cannot be used for this.
7) Line 445: it is not 9 classes of CLCs, but 9 members. They are not all channels, some are exchangers (as the authors themselves state).
8) Line 448: refs 208-210 don’t fit.
9) Line 454: CLCs are not the Cl channels important for regulatory volume decrease upon osmotic cell swelling. Those (LRRC8s) were identified in 2014 (Qiu et al, Cell; Voss et al, Science) and have since been confirmed by multiple labs in many functional and structural studies.
10) Line 456-457: see above comments on ref 20. This must be reviewed more critically. ClC-3 is not the VRAC (see also below, as this class of channels is lacking in the review).
11) An important class of ion channels that is lacking are volume-regulated anion channels (VRACs) that are formed by LRRC8 proteins (Qiu et al., 2014; Voss et al., 2014). They mediate uptake of certain cytostatics and are involved in apoptosis (Planells-Cases et al., 2015). In addition, there are conflicting reports on their roles in proliferation and cell migration in glioblastoma cells. Rubino et al., Front Oncology 2018 and Wong et al., J Cell Physiol 2018 propose a role in primary GBM, U251 and U87; however, Liu and Stauber, IJMS 2019 find no such effect by knockout in U251 and U87 cells.
Author Response
We thank the reviewer for the thorough and informative comments provided on the first version of the MS, and have substantially revised the MS to address all the concerns raised, as detailed below:
1) There is quite an overlap with a recent review by the same authors in the IJMS. The authors should make clear what the new direction of the present review is.
We agreed with this observation, and thank the reviewer for the candid fair assessments which have improved the quality of the MS. Components that overlapped with prior reviews have been removed or shortened to brief summaries, as appropriate for the flow of the presentation. The MS has been been refocused on specific ion channel classes relevant to GBM tumor cell motility.
2) Why Table 1A and 1B instead of Table 1 and Table 2?
Tables 1A and 1B have been renamed to Tables 1 and 2 respectively.
3) Lines 308f: “Transcriptomic analyses have identified classes of ion channels enriched in human 308 glioblastoma biopsy samples [79,154]” The authors should cite original research on the transcriptomics here rather than their own previous review/hypothesis paper – this is a general concern about the review: it often cites other reviews instead of the actual studies that showed (or did not) what the authors state.
The specified references have been revised to attribute development of The Human Protein Atlas to Uhlen et al. (2005 and 2012). The remainder of the review has been edited to ensure that the majority of the references cited are experimental papers, and include citations of work for both supporting and opposing arguments.
4) Line 311: There is more than voltage- and ligand-gated channels that are investigated, e.g. acid-sensitive (protonation is not a ligand), mechanosensitive, volume-regulated. These are not studied in respect to glioblastoma?
The sections on ion channels have been thoroughly edited to clarify the classes of channels involved, and expanded to cover evidence for roles of acid-sensing, mechanosensitive and volume-regulated channels.
5) Table 1B: what does ClC mean? Ref 20 is on ClC-3, which by now is known to be an endosomal Cl/H exchanger, no Cl channel of the plasma membrane. Therefore, unlike stated at various places in this review, ClC-3 cannot mediate Cl efflux for cell shrinkage. Instead, although ClC-3 briefly was a candidate for the volume-regulated anion channel, its identity has been revealed as LRRC8 proteins (see below). Ref 163 does not show anything on ClC. This part on chloride channels and exchangers is very superficial and unclear. Which CLC was actually shown to play what role in which process?
We appreciated the informative points raised by the reviewer. The outdated discussion of ClC-3 as a mediator of Cl– efflux for cell shrinkage has been removed. A new section on volume-regulated anion channels has been added that includes discussion of LRRC8 proteins.
6) Figure 4D does not make sense, the ion and water flow are mixed up. Water efflux does not parallel ion influx during RVD (possibly Cl in neurons, but then there needs to be efflux of other osmolytes that drives water efflux). Again, ref. 20 cannot be used for this.
We agree that the prior version of this MS included an overly general overview of voltage-gated ion channels, overlapping in part with concepts addressed in a previous review. The former Figure 4 and the corresponding text have been removed from the revised MS. The new version has been edited to focus more concisely on the ion channel classes that are emerging as attractive candidate targets for the development of novel GBM therapeutics.
7) Line 445: it is not 9 classes of CLCs, but 9 members. They are not all channels, some are exchangers (as the authors themselves state).
The discussion of ClCs has been removed from this review.
8) Line 448: refs 208-210 don’t fit.
We regret this error, which has been addressed.
9) Line 454: CLCs are not the Cl channels important for regulatory volume decrease upon osmotic cell swelling. Those (LRRC8s) were identified in 2014 (Qiu et al, Cell; Voss et al, Science) and have since been confirmed by multiple labs in many functional and structural studies.
New discussion of LRRC8s and VRACs have been added to this review in the context of regulation of cell volume by osmotic gradient control. Out-of-date discussions pertaining to functions of ClCs as regulators of cell volume have been removed.
10) Line 456-457: see above comments on ref 20. This must be reviewed more critically. ClC-3 is not the VRAC (see also below, as this class of channels is lacking in the review). 11) An important class of ion channels that is lacking are volume-regulated anion channels (VRACs) that are formed by LRRC8 proteins (Qiu et al., 2014; Voss et al., 2014). They mediate uptake of certain cytostatics and are involved in apoptosis (Planells-Cases et al., 2015). In addition, there are conflicting reports on their roles in proliferation and cell migration in glioblastoma cells. Rubino et al., Front Oncology 2018 and Wong et al., J Cell Physiol 2018 propose a role in primary GBM, U251 and U87; however, Liu and Stauber, IJMS 2019 find no such effect by knockout in U251 and U87 cells.
We appreciated the expert advice in the reviewer's points 10 and 11, and have removed discussion of ClC-3 in volume regulation. Building on the suggested references, we have added interesting details on VRACs and the roles of LRRC8 proteins.
Round 2
Reviewer 3 Report
The authors have addressed all concerns and clearly strengthened and updated the review during the revision.